## [Decision Letter · Decision Letter 0]

20 Aug 2025

PCOMPBIOL-D-25-01079

A novel transformer-based platform for the prediction and design of biosynthetic gene clusters for (un)natural products

PLOS Computational Biology

Dear Dr. Umemura,

Thank you for submitting your manuscript to PLOS Computational Biology. After careful consideration, we feel that it has merit but does not fully meet PLOS Computational Biology's publication criteria as it currently stands. Therefore, we invite you to submit a revised version of the manuscript that addresses the points raised during the review process.

Please submit your revised manuscript within 60 days Oct 20 2025 11:59PM. If you will need more time than this to complete your revisions, please reply to this message or contact the journal office at ploscompbiol@plos.org. Please include the following items when submitting your revised manuscript:

We look forward to receiving your revised manuscript.

Kind regards,

Boyang Ji, Ph.D.

Academic Editor

PLOS Computational Biology

Ilya Ioshikhes

Section Editor

PLOS Computational Biology

**Additional Editor Comments:**

Dear Dr. Umemura, and co-authors,

Thank you for submitting your manuscript and patience while awaiting peer review. In your study, it presented a transformer-based framework for the prediction and design of BGCs by leveraging a RoBERTa language model architecture. Reviewers recognized the potential importance and utility of this work. However, reviewers had raised substantive concerns that need be addressed before the manuscript can be considered further.

A few of major concerns - aggregated from reviewers comments below - are:

1. the redundancy between training and testing datasets

2. Insufficient details of hyper-parameter optimization

3. Lack of comprehensive performance benchmarking

4. reproducibility (the deposition of source code and documentation)

**Journal Requirements:**

3) Please amend your detailed Financial Disclosure statement. This is published with the article. It must therefore be completed in full sentences and contain the exact wording you wish to be published.

2) If any authors received a salary from any of your funders, please state which authors and which funders..

**Reviewers' comments:**

Reviewer's Responses to Questions

Reviewer #1: This study touches an important topic on biosynthetic gene clusters (BGCs). It presents a novel transformer-based framework for the prediction and design of BGCs, leveraging a RoBERTa language model architecture that treats functional domains as linguistic tokens. By training models on a range of genomic datasets, the authors demonstrate the model’s ability to capture domain context and suggest previously uncharacterized domain combinations. A strength of the work is the experimental validation of a model-predicted domain (FAD_binding_3) in the context of the cyclooctatin biosynthetic pathway, which has implications in natural product discovery and synthetic biology. The work is overall solid, but several points should be addressed to improve the clarity.

1. I have a few concerns regarding potential biases in the training data.

First, there appears to be a risk of data redundancy, as the BGC and genome datasets may contain many highly similar or near-identical sequences. Such redundancy could lead to model overfitting and reduce the validity of performance evaluations. I encourage the authors to assess and, if necessary, mitigate this redundancy.

Second, class imbalance across BGC types in the training data may also significantly affect model performance, particularly in tasks such as BGC type classification. Providing an overview of the distribution of BGC types used during training would help readers better interpret the reported results across different BGC classes.

2. For reproducibility and to better assess model robustness, I recommend that the authors include a description of the hyperparameter search strategy, including the search space explored (e.g., number of layers, attention heads, learning rate) and the criteria used for selecting the final model configuration.

3. On page 3, the manuscript states that “In prokaryotes, multiple genes in a BGC often form operons regulated by one or a few promoters,” and that “In contrast, eukaryotes lack operon structures, but BGC gene expression is frequently coordinated by transcription factors encoded within the cluster.” While these are broadly accepted observations, I recommend that the authors provide appropriate references to support these statements.

4. On page 10, in the masked prediction task, the authors train their model using BGCs from the antiSMASH database and evaluate performance using BGCs from the MIBiG database. However, since MIBiG entries may have highly similar or even nearly identical counterparts within the antiSMASH dataset, there is a risk that the evaluation does not fully reflect the model's ability to generalize domain context beyond memorized patterns. To ensure a rigorous assessment of model performance, please consider to quantify and minimize redundancy between the training and evaluation datasets. This can be achieved by applying a BGC clustering tool such as BiG-SCAPE, which would allow the identification of highly similar clusters between the two datasets.

Minor comments:

1. Figure 1: Please correct the label ‘Anabaena sp.’ to ‘Anabaena sp.’

2. Figure 2: Please correct the final phase label from ‘Maksed’ to ‘Masked’.

3. Figure 6: Please correct ‘MIBIG’ to ‘MIBiG’ and make the Type I PKS subclasses distinguishable, as they are labeled with the same color.

4. Figure 10B: “the the extracts” in Fig. 10B caption should be corrected.

Reviewer #2: Thank you for the opportunity to review “A novel transformer-based platform for the prediction and design of biosynthetic gene clusters for (un)natural products”. In this manuscript, the authors created a model that predicts domain types based on their surrounding domain context. The authors created four models based on the same architecture but trained on different datasets. The authors then showcase the utility of these models by predicting BGC compound classes and predicting domain types of masked domains from MIBiG BGCs. Finally, the authors perform a case study in which they add an additional untyped domain to an existing BGC, predict its domain type, and engineer and express the novel BGC to create a new natural product compound. The manuscript explores the fascinating topic of synthetic biosynthetic gene clusters, and the authors provide a new, valuable computational tool to aid scientists in designing them. Overall, I found the manuscript to be an interesting read. However, some major concerns need to be addressed before it is ready for publication.

Major concerns:

1. The manuscript currently uses a purely random train/test split, which risks scattering exact or near-duplicate samples across partitions and allowing the model to “cheat” by memorizing rather than generalizing. This also undermines the authors’ claim that the models understand domain context to any degree (lines 224-225). To mitigate memorization, I recommend the authors to (1) cluster by similarity and then perform a stratified train/test split so that duplicates and near-duplicates reside in the same set, (2) report both training and validation losses (Fig. 4 shows only validation), (3) check for data leakage by measuring n-gram overlap between generated output (i.e., for the case study) and the training corpus to catch any verbatim reproductions. Implementing these steps will clarify whether the authors’ models truly learn underlying principles or regurgitate their training data. For clustering based on the similarity of BGCs, the authors could potentially use BiG-SCAPE or BiG-SLiCE. For clustering based on genome similarity, the authors can utilize phylogenetic trees. Additionally, I recommend that the authors do 5-fold cross-validation based on their 20% validation test split and report on any (potential) differences in performance between the models trained on different folds. This will give the reader insight into the models’ generalizability.

2. The manuscript currently lacks a simple baseline for BGC compound class classification and offers no clarity on how hybrid BGCs are labeled. I recommend that the authors (1) introduce a baseline classifier (e.g., a shallow decision tree) to benchmark the model’s performance on distinguishing compound classes. I expect compound classes, such as T1PKS and NRPS, to be easily classifiable. Also, I recommend that the authors (2) clarify their current compound class labeling by providing examples of hybrid BGCs and specifying how classes were assigned to them. For example, a quick look at the MIBiG repository overview shows many BGCs with multiple compound classes assigned to them. I recommend that the authors reformulate this classification problem as a multi-class classification problem. I understand that this might be outside of the capabilities of the current model, in which case the analysis might need to be omitted.

3. Consequently (following up on major concern 2), I suspect misclassifications in Figure 6 might be attributed to the multi-class compound class reality of BGCs. This can be investigated by including a confusion matrix of the classification performance, instead of merely showing the accuracy for each class. The confusion matrix would also give insight into the sizes of each class, as I expect the classes to be highly unbalanced.

4. Although model performances for functional domain prediction appear impressive, the results are currently impossible to contextualize. To understand how the models understand BGC architectures, it would be clarifying if the authors include analyses on which domain types are generally predicted correctly (top 1), and which are usually not predicted correctly (top >100). To help interpret the results, I recommend that (1) the authors report counts per found domain type in both the training and test sets and (2) correlate frequency with prediction accuracy (common, predictable domains will likely score better than rare ones). Additionally, if the correct domain is in the top 10, what are the other top predicted domains? A model that understands BGC architectures should only predict similar or logical domain types in the top 10 compared to the correct domain type. I recommend that the authors at least check a few examples manually and include these examples in the manuscript.

5. The authors rightfully observe that the predictions in Table 3, as presented by Model I and Model IV, share similarities but also significant differences. I recommend that the authors include the rank of each top-predicted domain in one model in the predictions of the other model and provide a possible explanation for why the predictions differ. Additionally, the authors should provide a rationale for why the predictions of Model IV were used for experimental validation rather than those of Model I.

Minor concerns:

1. I suggest the authors upload the code used for training, validation, and plotting on GitHub (or a similar platform), in addition to Zenodo. GitHub allows users to track development (which would be in line with the authors’ vision to explore further the technology developed for this manuscript), potential collaboration, and issue tracking. A snapshot of GitHub can be easily downloaded and archived on Zenodo for every publication.

2. The Zenodo folder (and ideally also GitHub) should contain a README and/or an explanation of what is included in every file and repository uploaded. This is currently unclear. It should be clear from the README and/or explanatory file how to perform the analyses that provide the results for plots displayed in the file. This is necessary to ensure the analyses performed in the manuscript are reproducible.

3. Lines 156-158: Please clarify what “positioned centrally” means, as Dataset I appears to have the most samples and the fewest domain tokens (Table 1).

4. Table 1 lists Dataset III (bacterial genomes), comprising 9,748 strains, and Dataset IV (bacterial and fungal genomes), comprising 11,884 strains (2,136 additional genomes), for training the transformer-based models. This is 80% of the collected genomes. I advise adding this explicitly to the caption of the table.

5. Figure 1b: Please remove the gray triangle in the background, as it doesn’t seem to annotate anything in particular.

6. Figure 5: The figure was constructed based on the results of the actinomycetes BGCs dataset (Dataset II). Why was this model used and not any of the other models trained on Datasets I, III, and IV? I’d be especially interested to see the results of the model trained on Dataset I, which contains many more BGCs.

7. Figure 6: How were compound classes assigned for BGCs that have multiple associated compound classes (also see: major concern 2)?

Reviewer #3: This study introduces a transformer-based framework for predicting and designing biosynthetic gene clusters (BGCs), marking a significant methodological advancement over traditional tools like antiSMASH, which rely heavily on known domains and struggle to uncover novel structures. The research spans the genomes of bacteria, actinomycetes, and fungi, covering a variety of natural products and demonstrating the framework's reliability and broad applicability. To validate its practicality, the authors successfully expressed one of the predicted domains in Streptomyces albus, resulting in the discovery of an unknown cyclooctatin derivative. LC-MS analysis revealed that this compound shares the same molecular formula as cyclooctat-9-ENe-5,7-diol but exhibits different retention times, suggesting it is a novel structural isomer, further highlighting the tool's innovation and utility. By employing a closed-loop process that integrates AI-driven prediction, natural template screening, and heterologous expression, the framework enables domain mining and functional verification beyond traditional BGCs, offering a new paradigm for understanding biosynthetic mechanisms and enabling targeted modifications of natural products. This approach holds considerable potential for advancing drug development and synthetic biology.

Major comments

1. Verification cases need to be expanded: Currently, only a BGC of cyclooctatin from Streptomyces sp. ISL86 is used as the validation model. Given that the tool covers a wide range of species and natural product types, supplementary tests should be conducted in BGC with a longer evolutionary distance to confirm its universality.

2. Structural analysis to be improved: Due to the low quantity of the product, the structural confirmation has not been completed. The product structure needs to be analyzed by optimizing the expression conditions or using high-sensitivity detection techniques to clarify the functional mechanism of the new domain.

3. The rationale for selecting “0.7 accuracy threshold” (Line 199) should be clarified and justified? Was this benchmark based on prior literature, the baseline performance of existing tools, or a specific biological application requirement? Additionally, regarding the 0 accuracy observed for certain BGC classes, the authors should clarify and discuss whether this reflects a complete failure of model learning in these categories or potential data limitations? How this issue may affect the results and possible solutions should also be properly discussed.

4. For adopters in synthetic biology, a side-by-side comparison with widely used tools is critical to evaluate real-world utility. Quantifying such practical advantages would amplify the method’s impact.

Minor comments:

Table 1: clarify and provide rationales why these four datasets are co-selected and evaluated.

Fig. 5 : add the missing Y-axis label

The Codes, a dataset and models: should be briefly introduced in the main text, especially how readers may utilize or develop them.

**Have the authors made all data and (if applicable) computational code underlying the findings in their manuscript fully available?**

Reviewer #1: Yes

Reviewer #2: Yes

Reviewer #3: Yes

PLOS authors have the option to publish the peer review history of their article (what does this mean? ). If published, this will include your full peer review and any attached files.

**Do you want your identity to be public for this peer review?** For information about this choice, including consent withdrawal, please see our Privacy Policy .

Reviewer #1: No

Reviewer #2: No

Reviewer #3: No

**Figure resubmission:**
---

## [Decision Letter · Decision Letter 1]

9 Feb 2026

Dear Prof. Umemura,

We are pleased to inform you that your manuscript 'A novel transformer-based platform for the prediction and design of biosynthetic gene clusters for (un)natural products' has been provisionally accepted for publication in PLOS Computational Biology.

Best regards,

Boyang Ji, Ph.D.

Academic Editor

PLOS Computational Biology

Ilya Ioshikhes

Section Editor

PLOS Computational Biology

Reviewer's Responses to Questions

**Comments to the Authors:**

Reviewer #1: The authors have addressed the reviewer comments sufficiently, and I think this manuscript is ready for publication.

Reviewer #2: I appreciate the author's detailed responses and the improvements made to the manuscript. The authors have addressed most of my concerns satisfactorily. I have no further major comments.

Reviewer #3: The authors have addressed my comments.

**Have the authors made all data and (if applicable) computational code underlying the findings in their manuscript fully available?**

Reviewer #1: None

Reviewer #2: Yes

Reviewer #3: Yes

PLOS authors have the option to publish the peer review history of their article (what does this mean? ). If published, this will include your full peer review and any attached files.

**Do you want your identity to be public for this peer review?** For information about this choice, including consent withdrawal, please see our Privacy Policy .

Reviewer #1: No

Reviewer #2: No

Reviewer #3: **Yes:** Yes

---

## [Editor Report · Acceptance letter]

PCOMPBIOL-D-25-01079R1

A novel transformer-based platform for the prediction and design of biosynthetic gene clusters for (un)natural products

Dear Dr Umemura,

I am pleased to inform you that your manuscript has been formally accepted for publication in PLOS Computational Biology. Your manuscript is now with our production department and you will be notified of the publication date in due course.

With kind regards,

Anita Estes
